# Comparative Analysis of the Effect of Carbon- and Titanium-Ions Irradiation on Morpho-Anatomical and Biochemical Traits of *Dolichos melanophthalmus* DC. Seedlings Aimed to Space Exploration

**DOI:** 10.3390/plants10112272

**Published:** 2021-10-23

**Authors:** Veronica De Micco, Sara De Francesco, Chiara Amitrano, Carmen Arena

**Affiliations:** 1Department of Agricultural Sciences, University of Naples Federico II, Via Università 100, 80055 Portici, Italy; sara.defrancesco@unina.it (S.D.F.); chiara.amitrano@unina.it (C.A.); 2Department of Biology, University of Naples Federico II, Via Cinthia, 80126 Napoli, Italy; c.arena@unina.it; 3Interuniversity Center for Studies on Bioinspired Agro-Environmental Technology (BAT Center), 80055 Portici, Italy

**Keywords:** bioregenerative life support systems (BLSSs), heavy ions, ionizing radiation, morpho-anatomical traits, plant radio-resistance, space exploration, space food

## Abstract

The realization of manned missions for space exploration requires the development of Bioregenerative Life Support Systems (BLSSs) to make human colonies self-sufficient in terms of resources. Indeed, in these systems, plants contribute to resource regeneration and food production. However, the cultivation of plants in space is influenced by ionizing radiation which can have positive, null, or negative effects on plant growth depending on intrinsic and environmental/cultivation factors. The aim of this study was to analyze the effect of high-LET (Linear Energy Transfer) ionizing radiation on seed germination and seedling development in eye bean. Dry seeds of *Dolichos melanophthalmus* DC. (eye bean) were irradiated with two doses (1 and 10 Gy) of C- and Ti-ions. Seedlings from irradiated seeds were compared with non-irradiated controls in terms of morpho-anatomical and biochemical traits. Results showed that the responses of eye bean plants to radiation are dose-specific and dependent on the type of ion. The information obtained from this study will be useful for evaluating the radio-resistance of eye bean seedlings, for their possible cultivation and utilization as food supplement in space environments.

## 1. Introduction

The human exploration of Mars represents one of the most ambitious challenges that man will face in the coming years [1]. To realize long-duration manned missions, numerous obstacles must be overcome, regarding both organism’s adaptation to extreme environmental conditions and technical/operational issues [2,3]. Currently, the re-supply of resources is still an open issue, as for short-duration missions supplies are entirely shipped from Earth. This is clearly unfeasible for long-term manned missions, where resources must be regenerated directly onboard in the Bioregenerative Life Support Systems (BLSSs) to make the crews self-sufficient. In BLSSs, plants can be used to regenerate the air through the photosynthetic process, purify water by transpiration, recycle part of the crew waste products, produce fresh food on board, and help to create an Earth-like environment to mitigate the astronauts’ psychological stress due to the isolation conditions [4,5].

However, the efficiency of plants as regenerators can be influenced by space environmental factors affecting plant growth and metabolic processes [3]. Even though the type and level of stressors encountered in the different mission scenarios (e.g., space stations, and Lunar and Martian surfaces) are variable, there is common agreement that ionizing radiation (IR) risk represents a major constraint to human exploration of space, being radiation responsible for aberrations, both in mammalian and plant cells [6,7,8,9,10]. Specifically, outside the Low Earth Orbit (LEO), all organisms are exposed to: (i) chronic low-doses of galactic cosmic rays (GCRs), mainly composed of high-energy protons, alpha particles and heavy ions (HZE—high-energy nuclei component); and (ii) solar energy particles (SEPs), consisting in the short term of medium-low energy protons and alpha particles [10,11,12,13]. Considering the reduced possibility to expose plants to chronic radiation (e.g., limited access to space facilities and reduced availability of ground-based sources to simulate a space chronic radiation), experiments with specific ions at given acute doses are considered a required preliminary step in space biology to explore the radio-resistance of different species and evaluate their suitability for cultivation in space [3]. Recent research has reported that the effects of a chronic exposure are more severe compared to acute irradiation in wheat and Arabidopsis [14,15]. However, radiation effects on plants strictly depends on the plant species and the absorbed doses; therefore, as the effects of IR on plants are understood best at acute high doses [3], experiments based on acute exposure can help identify target species with very low radio-resistance at already low doses. Carbon and titanium are among ions considered proxy to galactic nuclei and are most commonly used to simulate the GRC spectrum in irradiation facilities on Earth [16,17]. Indeed, a deep understanding of the acute effect of these ions on plants, furnishes a base to assess the radio-resistance of a given species, before designing experiments on the chronic effects of HZE radiation that will constraint plant growth in during long-term missions.

On a biological level, IR is responsible for radiolysis processes, involving water and other chemical compounds leading to the formation of reactive oxygen species (ROS) [18,19,20]. ROS, such as superoxide radicals, hydroxyl radicals, and hydrogen peroxide, can disrupt cellular redox homeostasis, inducing an oxidative stress leading to a destructive process, such as lipid peroxidation or oxidative modification of proteins and nucleic acids, with serious consequences on cellular life and activity [21,22]. Although plants have proved to be more tolerant to ionizing radiation than animals, mainly due to the presence of mechanical barriers (e.g., specialized and thickened cell walls, cuticle, and pubescence), and the major level of ploidy shown by some species [23,24], their responses to radiation may vary depending on the quality (high or low Linear Energy Transfer—LET), dose, type of exposure (acute or chronic), species and cultivar, and phenological stage [7], and may involve genetic, metabolic, and morpho-anatomical modifications [25,26]. Generally, high doses (>100 Gy for seeds; >50–70 Gy for vegetative stages) [22] induce much more detrimental effects than low doses [22], at which null or positive effects have been reported [7,27]. At the same dose, high-LET (Linear Energy Transfer) radiation (e.g., protons and heavy ions) has a lower ability to penetrate through seed teguments and to cross plant cell layers, but has a higher mutagenic action than low-LET radiation (X and γ rays) [28,29,30]. The analysis of the hit effect of HZE (including carbon and titanium ions) needs to be explored when evaluating space radiation risks for long-term manned missions [10], as it could offer a more realistic scenario of GCR than low-LET radiation. Although it is commonly accepted that the plant response is ion-specific, most available information regards a few types of ions with carbon-ion beam (CIB) irradiation among the most widely used, due to its common applications in mutation breeding programs. For this ion, in several plant species, dose-dependent responses as well as stimulating effects in seedling growth have been reported [31,32]. For instance, in *Solanum lycopersicum* L. ‘Microtom’, high-LET radiation at specific doses has increased the photochemical efficiency, the amounts of D1 protein, and photosynthetic pigment content [33]. Moreover, in the same study, adult plants developed from irradiated seeds produced larger tomato fruits, richer in antioxidants (carotenoids, anthocyanins, and ascorbic acid) compared to non-irradiated controls. 

Even when irradiation does not affect germination percentage and rate, there is interest in studying the post-germination effects for two main reasons: (i) seedling development is one of the most delicate processes in plant life cycle, and (ii) deep knowledge on possible alterations induced by radiation is needed to design strategies to facilitate their establishment in space growth chambers [34]. Moreover, although not having a role in the resource regeneration, seedlings are interesting candidates as food complements to crew’s diet, as they are easy to produce and highly nutritious in terms of antioxidants, minerals, and vitamins [35,36].

Normally, until seedlings reach the condition of photo-autotrophy, their development in the post-germination phase is essentially based on the reserves stored in seeds, such as carbohydrates, proteins, and lipids [37]. Among them, starch is the major carbohydrate storage in plants, which plays a key role in helping plants in the post-germination phase reacting to abiotic stress. In fact, under stressful conditions, plants generally re-mobilize starch to provide energy and carbon when photosynthesis may be potentially limited, allowing the plant to grow and stabilize [38]. It is known that the accumulation of ROS due to exposure to high doses of ionizing radiation can interfere with structural and functional organic molecules, causing disturbance to the cellular metabolism [39,40], which may in turn compromise the efficiency of reserve mobilization and therefore the survival of the seedlings.

In the present study, dry seeds of *Dolichos melanophthalmus* DC. (eye bean) were exposed to C-ions and Ti-ions at two doses to evaluate the effect of different sources of radiation on seed germination and seedling growth in terms of morpho-anatomical and biochemical traits. The eye bean, also known as *Vigna sesquipedalis* L., was chosen due to the high nutritional content of its sprouts, rich in proteins and essential ammino-acids. Ti-ion was chosen as, although it is considered to simulate GCR well, there is not much information on its acute effects on plants, while C-ion was considered as a reference radiation as it is widely used in experiments on plant biology. The doses utilized in our experiment, namely 1 Gy and 10 Gy, are considered low for plant organisms, and were specifically chosen to avoid detrimental outcomes and induce stimulatory effects on plants [3,33,41]. Therefore, the knowledge gained from this study may be useful to achieve a first evaluation of the radiation-induced morpho-anatomical and biochemical responses of eye bean seedlings, in order to include data on radio-resistance in the decision process for: (i) evaluating whether eye bean is a good candidate for food supplement in long term manned space missions in which plants would be exposed to chronic radiation; and (ii) defining cultivation requirements in BLSSs.

## 2. Results

### 2.1. Germination Rate and Seedling Length

Both control and Ti-irradiated seeds at both doses showed a survival rate of 100% (Figure 1a). On the contrary, the C-irradiated seeds showed a significant decrease at the doses of 1 Gy (15%) and at 10 Gy (20%). Seedling length (including root and hypocotyl) was significantly reduced in the 10C seedlings compared to the other treatments (Figure 1b).

### 2.2. Anatomy

Microscopy analysis showed that seedlings originated from the seeds irradiated with both ions at the two doses maintained the normal structure in cotyledons and hypocotyls, with no evident qualitative alterations (Figure 2). The quantification of anatomical traits evidenced significant effects, especially at the 10 Gy dose, with both ion types. Regarding cotyledons, 10C and 10Ti seedlings showed larger cells when compared with the other treatments and control, as indicated by significantly higher values of cell area and Feret diameters (Figure 2, Table 1). Cell elongation was not significantly influenced by irradiation, as shown by the similar values of aspect ratio and sphericity in all treatments (Figure 2, Table 1). Irradiation at the lower dose of both ions determined a cell shape with significantly lower values of convexity in 1C and 1Ti seedlings compared to non-irradiated control; 10C and 10Ti showed intermediate values (Table 1). Irradiation also elicited significant effects on starch percentage, and number and size of amyloplasts (Figure 3, Table 1). Except for 1C, seedlings from irradiated seeds showed a significantly lower starch percentage than control (Table 1). The lower percentage of starch in cotyledons was due to significantly lower number of smaller amyloplasts per cell (Table 1). In 1C, the reduced amyloplasts size was compensated by their higher number per cell which ultimately determined the similar starch percentage compared to non-irradiated control (Table 1).

As regards hypocotyl, significant variations induced by irradiation were observed in the cortical cylinder (Table 2). The parenchyma cell size was significantly lower in 1Ti and 10C seedlings which also showed significantly lower values of convexity if compared with the other treatments. As regards cell shape, 1C and 10Ti seedlings tended to have more elongated cells as shown by, respectively, higher and lower aspect ratios and sphericity compared to the other treatments. Considering the starch content and distribution, seedlings from seeds irradiated with Ti showed a significantly higher starch percent per cell than C-irradiated and controls, with maximum values at the highest dose (Figure 3, Table 2). The higher content of starch was related to a higher number of larger amyloplasts compared to the other treatments. This trend was consistent with the distribution of starch among different cortical cell layers (Table 2). Higher values of the ratio between the portion of tissue containing amyloplasts and the total thickness of the cortical cylinder (TA/TTCC), were also observed in 1Ti and 10Ti apart from 1C seedlings (Table 2). The intercellular spaces did not show significant variations in terms of shape and size among treatments (Table 2).

Regarding parenchyma cell size in the stele, only the Feret maximum diameter was significantly higher in 10Ti than the other treatments, with 10C seedlings showing intermediate values (Table 3). The higher Feret maximum diameter in 10Ti suggested a more elongated cell shape in 10Ti seedlings as confirmed by the low values of sphericity. As regards the parameters related to starch, trends of variation were very similar to those found in the cortical cylinder. Indeed, 1Ti seedlings showed the highest values of starch percent, which was significantly higher than in 10Ti, which in turn was characterized by significantly higher values than the other treatments (Figure 3, Table 3). Furthermore, in this tissue, the higher values of starch percent in seedlings from Ti-irradiated seeds were due to both a higher number of larger amyloplasts and a more extended portion of tissue layers characterized by starch accumulation (Table 3). As in the cortical cylinder (Table 2), in the stele there were no significant differences in size and shape of the intercellular spaces (Table 3).

### 2.3. Biochemical Traits

The concentration of H_2_O_2_ was affected by irradiation which determined a significant increase only at the highest dose (10 Gy) in both ions (Figure 4a). The same trends were found for ascorbic acid (AsA) content which reached the highest value after irradiation, as well at 10 Gy (Figure 4b).

The photosynthetic pigment content, namely chlorophylls a, b, and a + b, and carotenoids x + c, (Table 4), showed significant variations in response to ionizing radiation. Compared to control, a decrease in chl a, total chlorophyll (chl a + b), and total carotenoids (x + c) was observed in 10C seedlings. Regarding Chl b, only seedlings sprouted by 10C seeds showed a significant reduction compared to control and other treatments.

## 3. Discussion

This study highlighted the role of ionizing radiation in influencing seedlings’ morpho-anatomical development and nutritional quality. When subjected to different doses of carbon and titanium ions, eye bean showed a different capacity of development and morpho-anatomical acclimation, accompanied by changes in biochemical traits, ultimately resulting in a modified nutritional content. 

Germination is a delicate process, and the first stage of development, characterized by the seedling establishment, may be inhibited by a multitude of abiotic factors such as drought, light, salinity, pH, and temperature [42]. To date, compared to other environmental factors, the effect of ionizing radiation has been less explored; however, contrasting results have been found depending on the type of radiation, dose, and time of exposure. Just to mention a few studies [36,43], no reduction in the germination percentage has been reported in seeds of *Solanum lycopersicum* L. and of *Vigna radiata* L. irradiated with increasing X-rays doses (up to 50 Gy). Differently, when seeds are irradiated with high-LET radiation, reduction in germination percentage and survival can occur at specific doses, as in the case of *Nicotiana tabacum* L. seeds irradiated with carbon ions inducing chromosome aberrations [44]. In tomato, irradiation of dry seeds with Ca ions at 25 Gy has also been reported to reduce the germination rate [33]. Our results are partially in agreement with previous findings, in so far as irradiation with C-ion at both doses (1 and 10 Gy) was responsible for a reduction in the germination percentage. The reduced germination was accompanied by a decline in seedling final size only at the highest dose (10 Gy), suggesting that the 1 Gy dose is too low to prevent growth and only interferes with the early stages of seed germination. On the contrary, irradiation of eye bean seeds with Ti-ions at the tested doses did not influence growth, confirming that germination responses depend on the ion type [7]. Once the very early stages of seed germination and radicle emergence were overcome, it seemed that, although slowed in 10C treatment, eye bean seedling growth occurred without significant morpho-anatomical aberrations. The highest doses of both ions led to an increase in cell size in cotyledons but not in hypocotyls, confirming that the influence of radiation can vary among organs and tissues as found in other species [36,45]. The increased size of cells in cotyledons of irradiated seeds might have been favored by a possible radiation-induced loosening of cell walls, which may have reduced the constraint to the protoplast, given the high turgidity (i.e., high values of convexity), even if such cells were not actively growing. An increased size of cotyledon cells has also been reported in seedlings grown in space, and ascribed to space-induced anomalies in the development of cell walls (cell wall loosening due to degradation of specific components and altered cellulose microfibril distribution) which would not be able to constrain the enlargement of the protoplast [46,47]. Such a phenomenon is even more evident when developing organs are irradiated as a target stage, as in the case of leaves of *Phaseolus vulgaris* L. irradiated with X-rays, as during morphogenesis the mechanism by which cell wall loosening induces wall stress relaxation (which generates the reduced water potential that is needed for water uptake and cell expansion) occurs while cell wall material is still depositing, thus determining less mechanical constraint [40,48].

The irradiation with C-ions seemed to be more effective in inducing morpho-anatomical quantitative modifications if compared with Ti-ions irradiation and control. Moreover, the seedling response was dependent on the dose in the case of C-ions, with the 10C dose that can be considered already as a stressful value. In fact, seedlings from seeds irradiated with carbon at 10 Gy showed a decrease in cell size and turgor pressure, while at the dose of 1 Gy, the highest values of cell area and convexity were found. This variation of cell size and convexity due to carbon irradiation, especially at the highest dose, could also explain the minor values of the total thickness of the cortical cylinder and the length of the hypocotyl/radicle. Compared to the cortical cylinder, the stele appeared to be less sensitive to radiation, as if inner tissues containing the vital vascular tissue would be preserved and maintained more stable [49].

The quantification and localization of the starch in the different tissues of eye bean seedlings allowed us to interpret the dynamics of reserve mobilization in response to irradiation treatments. Ti-ions at both doses caused a quicker mobilization of starch from cotyledons to both cortical cylinder and stele in hypocotyls, especially at the lower dose compared to control and C-ion treatment. The latter induced such a mobilization only at the highest dose, and especially towards the cortical cylinder. Starch is a glucose homopolymer, deposited in amyloplasts, and represents the main storage of carbohydrates in plants [50]. Recent reviews have pointed out the role of starch in the abiotic stress tolerance [38,51]. Indeed, during the environmental stress events (e.g., drought, salinity, or temperature), when the assimilation of carbohydrates can be compromised, starch metabolisms act as a buffer and as a “carbohydrates-source” when carbon is necessary or as a “carbohydrates-sink” when sugars are in excess [52]. Little is known about the effect of ionizing radiation on starch metabolism and amyloplasts morphology; however, here, the accumulation of starch in hypocotyls suggests its possible mobilization to fuel the growth of the stem. This would further explain the greater length of the hypocotyls/radicle observed in seedlings from the Ti-irradiated seeds. The seedling reaction after the irradiation of seeds with C-ions was different. The reduction in starch accumulation in 10C cotyledons, accompanied by a moderate increase in its content only in the cortical cylinder, suggests a conversion of starch into sugars to compensate for the stressful condition induced by the highest dose at a growth stage when an adequate sugar supply is not already guaranteed by photosynthesis. Altered starch metabolisms have been recently reported in *Vigna radiata* L. subjected to environmental stress (high evaporative demand) where a decline in leaf starch content with reduction in net-photosynthesis were explained as a down-regulation of carbon metabolism triggered by the stress, which also lead to soluble carbohydrate mobilization, as also suggested in other species [35,36,53,54,55,56,57]. Moreover, in eye bean seedlings, the lower values of photosynthetic pigments (chlorophylls and carotenoids) in 10C seedlings compared to the other treatments may determine simultaneous reduction in the light-harvesting capacity and may consequently limit photosynthesis at later developmental stages (adult plants).

Although reduced growth was evident in the morphology of the sole 10C seedlings, the measured values of H_2_O_2_ and ascorbate suggested that both ions induced a stress condition in seedlings at the dose of 10 Gy. To avoid oxidative damage, plants may benefit from different non-enzymatic phyto-protectants, such as phenolic compounds, carotenoids, ascorbic acid (AsA), tocopherol, and glutathione, or enzymatic compounds, such as catalase (CAT), superoxide dismutase (SOD), glutathione reductase, and peroxidase (GR and GPx), which act as redox buffers and influence the expression of the genes involved in cell-protective response and defense pathways [40,58]. More specifically, increments in ascorbic acid have been found in many plant species subjected to abiotic stress (drought, salinity, temperature, or humidity) as a defense against oxidative stress [59,60,61,62]. As the human body cannot synthetize the ascorbic acid endogenously, it represents an essential nutriment, and sprouts rich in ascorbic acid can help in counteracting the negative effect of ionizing radiation and microgravity in astronauts, acting as natural radio-protectants [63]. Seedlings from seeds subjected to Ti-ion at 10 Gy, together with increments in ascorbate, also enhanced the carotenoid content compared to the 1 Gy dose, again reaching the levels of the control seedlings. Carotenoids, besides their function as photosynthetic pigments, are annumerated among these radio-protectant molecules, which have since been found to decrease the potential stress of ROS within the metabolism [64].

In conclusion, the overall analysis of morpho-anatomical and biochemical traits indicated that seedlings developed from dry seeds of eye bean irradiated with the two different ions, at the two doses, show different responses, highlighting that: (i) plant response to high-LET ionizing radiation is different if irradiated with carbon or titanium ions; (ii) dose-dependent responses can vary in the different analyzed traits and in different organs/tissues due to their different sensitivity; and (iii) the mechanisms of seedling response to cope with radiation is different for the two ions. In the case of C-ion, although there was a decrease in germination, survived seedlings did not show impairment in the general structure of hypocotyls and cotyledons, but slowed growth, likely due to an altered starch metabolism that was less efficiently mobilized towards growing hypocotyls, compared to seedlings from Ti-irradiated seeds. By contrast, seedlings from Ti-irradiated seeds showed an improved mobilization of starch towards actively growing tissues, likely as a successful response to stressful conditions confirmed by the increase in the antioxidant content. The overall results suggested that high-LET radiation may act by increasing plant traits as the antioxidant content, while still allowing a normal morpho-functional development at the early stages of plant development that are crucial for cultivation establishment in BLSSs. Such traits also indicate seedlings as a source of food supplement with good radiation-induced nutraceutical properties. Further studies are needed, using other species, ions, and a wide range of doses, to obtain a complete understanding of the mechanisms of responses to ionizing radiation and confirm a vision in which ionizing radiation in space has to be considered, no longer as a constraint, but as a sort of stimulant factor for the production of high-nutritious plant-derived food. Considering that chronic exposure to radiation has been demonstrated to be more effective in determining changes at biochemical rather than al morphological level in other species [14], further studies also using chronic exposure are desirable to confirm the fitness of this species in space, still maintaining a suitable morpho-functional development while increasing nutritional traits as the antioxidant content.

## 4. Materials and Methods

### 4.1. Experimental Design

Dry seeds of *Dolichos melanophthalmus* DC., also known as *Vigna sesquipedalis* L. (eye bean) were purchased from a local provider, shipped to Germany, and divided into three groups: (i) non-irradiated control seeds (Ct), (ii) seeds to be irradiated with carbon ions (C), and (iii) seeds to be irradiated with titanium ions (Ti) (the latter two at two different doses (1 Gy and 10 Gy)). The species was chosen due to the high nutritional content of its sprouts, rich in proteins and essential ammino-acids [65]. The irradiation was performed using a pencil beam in a spread-out Bragg peak (SOBP), in the heavy-ion synchrotron (SIS) at GSI Helmholtzzentrum fur Schwerionenforschung (Darmstadt, Germany). Dry seeds were irradiated with C-ions [ Isotope ^12^C; Energy: 120 MeV/u (monoenergetic); LET: 80 keV/µm; Dose rate 2 Gy·min^−1^; Doses: 1 and 10 Gy] and Ti-ions [Isotope ^50^Ti; Energy:1 GeV/u (monoenergetic); LET: 108 keV/µm; Dose rate 2 Gy·min^−1^; Doses of 1 and 10 Gy]. These doses, largely below the threshold for occurrence of DNA damage [66], were chosen as they likely do not induce mortality and help assessing possible stimulatory effects on plant development and starch production. After the treatment, irradiated and control seeds were placed in closed boxes (shaded from light) and transferred to the Department of Biology at the University of Naples Federico II (Naples, Italy). Throughout the transfers, both groups of seeds were subjected to the same operations under the same environmental conditions to avoid any bias due to different pre-germination conditions other than irradiation and prepared for the analyses summarized in Figure 5.

### 4.2. Seed Germination and Morphology

Three sets of 30 seeds per each treatment (control, Ct; C-ion irradiated at 1 Gy, 1C; C-ion irradiated at 10 Gy, 10C; Ti-ion irradiated at 1 Gy, 1Ti; and Ti-ion irradiated at 10 Gy, 10Ti) were placed into sterile Petri dishes on three layers of filter paper imbibed with distilled water and incubated in the dark at 20 °C. Germination percentage was determined: seeds were considered germinated when the emerging root was longer than the seed maximum diameter. Seedlings from irradiated and control seeds were transferred into 15 cm diameter pots, filled with peat-based compost (peat:soil, 1:1 *v*:*v*) and placed in a growth chamber under controlled conditions of temperature (25 ± 1°C), relative humidity (RH 60 ± 10%) and light (photosynthetic photon flux density, PPFD, 155 ± 5 µmol photons m^−2^s^−1^). After 10 days, seedling length (SL = root length + hypocotyl length) of all seedlings was measured. Healthy seedlings were collected from each treatment and preserved for microscopy (*n* = 6) and biochemical (*n* = 5) analyses.

### 4.3. Light Microscopy and Digital Image Analysis

The seedlings per each treatment were fixed in F.A.A. (formaldehyde 40%: glacial acetic acid: ethanol 50% in the volume ratio of 5:5:90). The seedlings were dissected under a microscope (SZX11, Olympus, Hamburg, Germany) to obtain subsamples of cotyledons (a cross section of 5 mm thickness in the median zone) and hypocotyl (5 mm length as well). The subsamples were dehydrated in ethanol series (up to 95% ethanol) and embedded in acrylic resin JB4 (Polysciences, Warrington, PA, USA). Cross sections (5 μm thick) were cut through a rotary microtome obtaining two series of sections from each sample that were stained either with 0.025% (m/v) Toluidine blue in a 0.1 M citrate buffer at pH 4 [67] or in IKI solution [68]. The stained sections were mounted with mineral oil for microscopy and analyzed under a light microscope with transmitted light (BX61, Olympus). Images were collected by means of a digital camera (XM50, Olympus) and analyzed through the AnalySIS 3.2 (Olympus) software program to quantify the cytological and anatomical traits. As regards cotyledons, the starch percentage accumulated was estimated by measuring the area occupied by the amyloplasts in a given area (in 3 regions per section). Moreover, the number of amyloplasts per cell was counted in 10 cells per section, and for each amyloplasts the major and minor axis were measured. Finally, to give a background about morphological characteristics of the cells, the cell size and shape were quantified (in 10 cells per section) through the following parameters: cell area, maximum, mean, and minimum Feret diameters (i.e., the measured distance between parallel lines tangential to the cell perimeter in the cross section), aspect ratio (maximum width/height ratio of a bounding rectangle for the cell, defining how it is elongated), sphericity (roundness of a particle: a spherical particle has a maximum value of 1), and convexity (the fraction of the cell area and the area of its convex surface: when a cell is turgid the convexity value tends to 1, while when it is shrunk the convexity value is lowered) [47,69]. Regarding the hypocotyls, the quantification of cytological and anatomical traits was performed distinguishing the cortical cylinder and stele. More specifically, the total radius of stele (TRS) and the total thickness of cortical cylinder (TTCC) were measured in 3 regions per section. The thickness of the layers of cells containing the amyloplasts (TA) in both cortical cylinder and stele was measured in 3 regions per section, and their ratios with total TRS and TTCC were also quantified. The amount of starch (starch percentage) was estimated by quantifying the percentage of tissue area occupied by the amyloplasts by measuring the surface occupied by the amyloplasts in a given area, in 3 regions per section. The number of amyloplasts per cell was also counted in 10 cells per sections, and for each amyloplasts the diameter was also measured. Finally, the size and shape of cell and intercellular spaces were quantified in 10 cells per section, similar to the process for cotyledons.

### 4.4. Biochemical Analyses

All biochemical analyses were determined on five different leaf samples for each treatment for both control and irradiated plants, at harvest. The endogenous hydrogen peroxide (H_2_O_2_) concentration was used as marker of oxidative stress. The quantification of H_2_O_2_ content was carried out by means of a colorimetric method [70]. Briefly, 500 mg of frozen powder from leaves were extracted with 5 mL of ice cold 0.1% trichloroacetic acid (TCA) and the mixture was then incubated for 15 min on ice and centrifuged at 10,000 rpm for 15 min at 4 °C. After, 500 µL of surnatant were added 500 µL phosphate buffer 10 mM (pH 7.0) and 1 mL of potassium iodide (1 M). The mixtures were then incubated in the dark for 40 min and H_2_O_2_ quantified at 525 nm by a spectrophotometer (UV-VIS Cary 100, Agilent Technologies, Santa Clara, CA, USA). The concentration was expressed in mmol g^−1^ FW.

The ascorbic acid (AsA) content was evaluated using the Ascorbic Acid Assay Kit (MAK074, Sigma-Aldrich, St. Louis, MO, USA) following the procedure reported in [71]. Briefly, 10 mg of sample was homogenized in 4 volumes of cold AsA buffer, centrifuged at 12,850 g for 10 min at 4 °C. The surnatant was mixed with AsA assay buffer to a final volume of 120 μL. In this assay, the AsA concentration was measured by a coupled enzyme reaction developing a colorimetric (570 nm) product equivalent to the amount of ascorbic acid contained into the sample. The concentration of ascorbic acid in the samples was referred to a standard curve and expressed in ng mL^−1^.

Total chlorophylls (Chl a, b, and a + b) and carotenoids (x + c) were determined according to Lichtenthaler et al. [72]. Pigments were extracted from samples using mortar and pestle in ice-cold (4 °C) 100% acetone and centrifuged at 3200 g for 5 min (Labofuge GL, Heraeus Sepatech, Hanau, Germany). The absorbance of supernatants was quantified by a spectrophotometer (UV-VIS Cary 100, Agilent Technologies, Santa Clara, CA, USA) at 470, 645, and 662 nm and pigment concentration expressed in mg g^−1^ (FW).

### 4.5. Data Elaboration

All results were subjected to one-way analysis of variance (ANOVA) using the SPSS statistical package (SPSS Inc., Chicago, IL, USA) and the Duncan multiple comparison test (*p* ≤ 0.05). Kolmogorov–Smirnov test was performed to check for normality. Percent data were transformed through arcsine function before statistical analysis.

## Figures and Tables

**Figure 1 plants-10-02272-f001:**
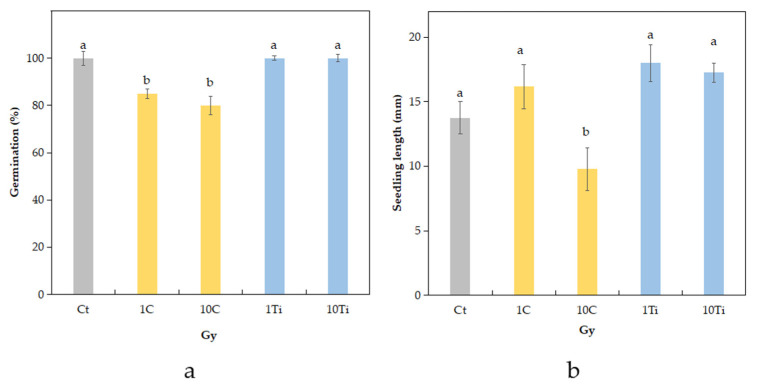
Germination percentage of eye bean seeds (**a**) and seedling length (**b**) of non-irradiated control (Ct) and seeds exposed to C- and Ti-ions at the doses of 1 and 10 Gy. Mean values and standard errors are shown (*n* = 30). Different letters correspond to significantly different values according to Duncan test (*p* ≤ 0.05).

**Figure 2 plants-10-02272-f002:**
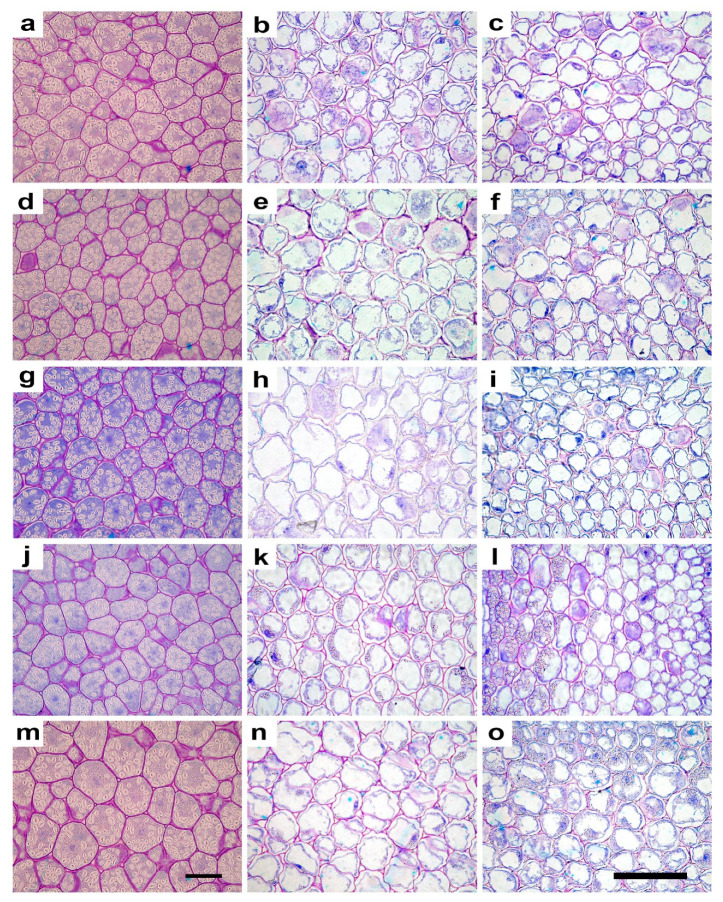
Light microscopy views of cross sections of cotyledons (**a**,**d**,**g**,**j**,**m**), hypocotyl stele (**b**,**e**,**h**,**k**,**n**) and hypocotyl cortical cylinder (**c**,**f**,**i**,**l**,**o**) in eye bean seedlings. Control (**a**–**c**) and irradiated seeds: C-ions at 1 Gy (**d**–**f**) and 10 Gy (**g**–**i**); Ti-ions at 1 Gy (**j**–**l**) and 10 Gy (**m**–**o**). Images (**a**,**d**,**g**,**j**,**m**) are all at the same magnification. Bars = 100 µm. Images (**b**,**e**,**h**,**k**,**n**,**c**,**f**,**i**,**l**,**o**) are all at the same magnification. Bars = 100 µm.

**Figure 3 plants-10-02272-f003:**
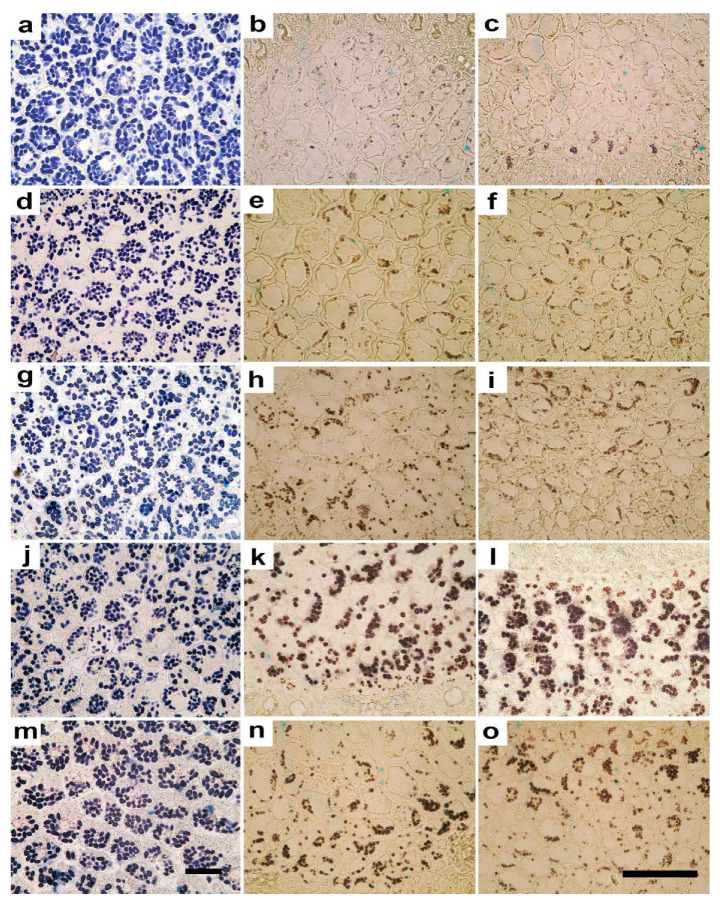
Light microscopy views of cross sections of cotyledons (**a**,**d**,**g**,**j**,**m**), hypocotyl stele (**b**,**e**,**h**,**k**,**n**) and hypocotyl cortical cylinder (**c**,**f**,**i**,**l**,**o**) in eye bean seedlings, showing amyloplasts in dark brown. Control (**a**–**c**) and irradiated seeds: C-ions at 1 Gy (**d**–**f**) and 10 Gy (**g**–**i**); Ti-ions at 1 Gy (**j**–**l**) and 10 Gy (**m**–**o**). Images (**a**,**d**,**g**,**j**,**m**) are all at the same magnification. Bars = 100 µm. Images (**b**,**e**,**h**,**k**,**n**,**c**,**f**,**i**,**l**,**o**) are all at the same magnification. Bars = 100 µm.

**Figure 4 plants-10-02272-f004:**
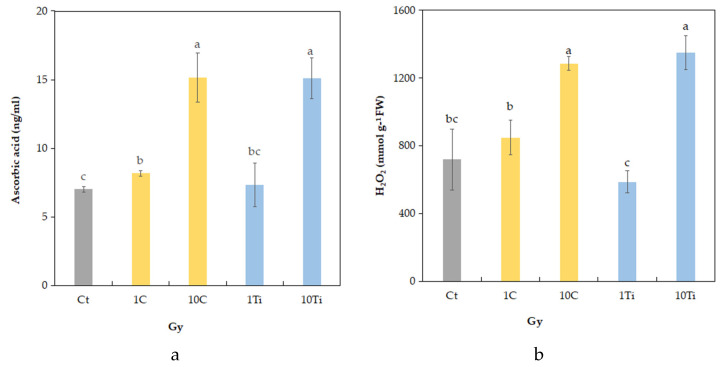
Content of H_2_O_2_ (**a**) and ascorbic acid (**b**) in seedlings of eye bean: comparison among non-irradiated control (Ct) and those exposed to C- and Ti-ions at the doses of 1 and 10 Gy. Mean values and standard errors are shown (*n* = 5). Different letters correspond to significantly different values according to Duncan multiple comparison tests (*p* ≤ 0.05).

**Figure 5 plants-10-02272-f005:**
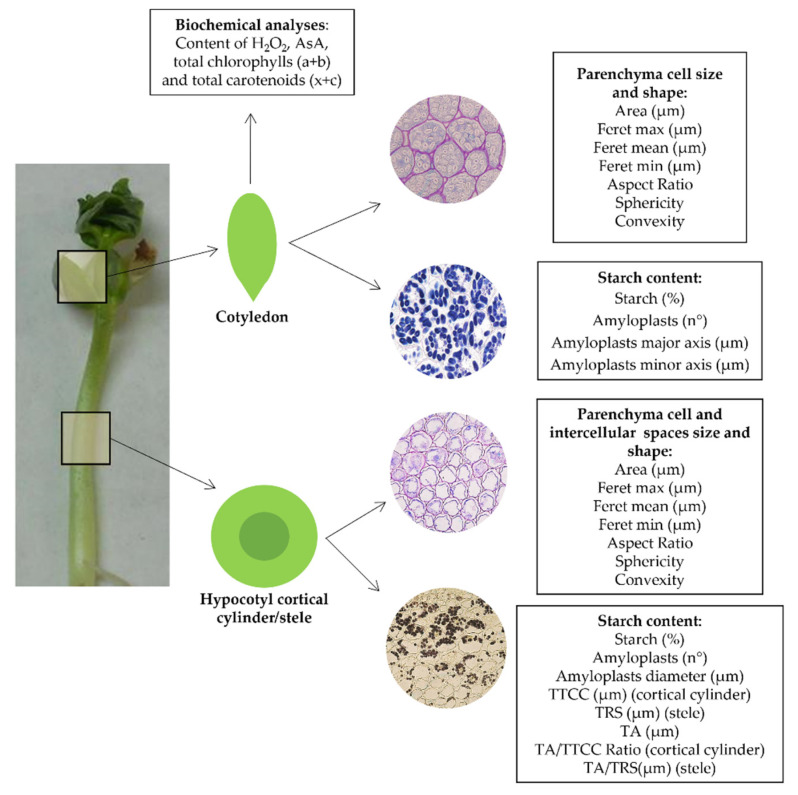
Diagram showing all the parameters measured divided for the different tissues.

**Table 1 plants-10-02272-t001:** Effect of irradiation on cell size (area and Feret maximum, mean, and minimum diameters), shape (aspect ratio, sphericity, and convexity), and starch content (starch%, number of amyloplasts, and amyloplast major and minor axes) in cotyledons of eye bean seedlings from control and seeds irradiated with C- and Ti- ions at 1 and 10 Gy doses. Mean values and standard errors are shown (*n* = 18 for starch%; *n* = 60 for all the other parameters).

Cotyledons						
	Ct	1C	10C	1Ti	10Ti	S
**Parenchyma Cells**						
Area (µm)	5243 ± 180.9 b	4631 ± 267.9 b	6740 ± 278.9 a	4947 ± 201.2 b	6928 ± 374.5 a	***
Feret max (µm)	99.68 ± 2.075 b	98.81 ± 3.413 b	113.0 ± 3.079 a	95.13 ± 2.205 b	112.5 ± 2.625 a	***
Feret mean (µm)	87.36 ± 1.491 b	84.56 ± 2.561 b	98.73 ± 2.295 a	84.94 ± 1.726 b	99.26 ± 2.425 a	***
Feret min (µm)	72.94 ± 1628 b	67.24 ± 1962 b	81.28 ± 2067 a	72.10 ± 1835 b	83.15 ± 2615 a	***
Aspect Ratio	1.341 ± 0.037 a	1.457 ± 0.041 a	1.385 ± 0.038 a	1.311 ± 0.033 a	1.363 ± 0.034 a	NS
Sphericity	0.595 ± 0.035 a	0.526 ± 0.034 a	0.550 ± 0.028 a	0.589 ± 0.032 a	0.571 ± 0.029 a	NS
Convexity	0.957 ± 0.007 a	0.904 ± 0.021 c	0.947 ± 0.007 ab	0.936 ± 0.020 b	0.952 ± 0.008 ab	***
Starch (%)	49.03 ± 0.009 a	46.66 ± 0.023 a	34.67 ± 0.010 b	35.46 ± 0.024 b	35.71 ± 0.011 b	***
Amyloplasts (n°)	20.78 ± 0.697 b	23.35 ± 0.962 a	20.28 ± 0.801 b	18.67 ± 0.914 b	20.58 ± 1024 b	*
Amyloplasts major axis (µm)	24.73 ± 0.556 a	20.57 ± 0.484 c	23.91 ± 0.450 ab	22.58 ± 0.594 b	23.05 ± 0.459 b	***
Amyloplasts minor axis (µm)	13.01 ± 0.316 a	11.05 ± 0.238 b	12.91 ± 0.244 a	11.70 ± 0.364 b	12.76 ± 0.330 a	***

NS, *; **, and ***—Not significant or significant at *p* < 0.05, 0.01, and 0.001, respectively. Different letters within each column indicate significant differences according to Duncan multiple comparison tests (*p* ≤ 0.05).

**Table 2 plants-10-02272-t002:** Effect of irradiation on cell and intercellular spaces size (area and maximum, mean, and minimum Feret diameters), shape (aspect ratio, sphericity, and convexity), and starch content (starch%, number of amyloplasts, amyloplast diameter, total thickness of cortical cylinder (TTCC), thickness of the layers of cells containing the amyloplasts (TA), and their ratio (TA/TTCC)) in hypocotyl cortical cylinder of eye bean seedlings from control and seeds irradiated with C- and Ti-ions at 1 and 10 Gy doses. Mean values and standard errors are shown (*n* = 60 for cells/amyloplasts/intercellular spaces; *n* = 18 for tissue thickness and starch%).

Hypocotyls—Cortical Cylinder						
Parameters	Ct	1C	10C	1Ti	10Ti	S
**Parenchyma Cells**						
Area (µm)	1086 ± 56.57 ab	1217 ± 87.22 a	665.2 ± 54.66 c	933.9 ± 80.77 b	1145 ± 110.6 ab	***
Feret max (µm)	43.96 ± 1.298 a	44.32 ± 1.427 a	35.79 ± 1.451 b	42.03 ± 2.124 a	43.27 ± 2.071 a	**
Feret mean (µm)	39.41 ± 1.027 a	40.81 ± 1.387 a	31.43 ± 1.213 b	36.57 ± 1.701 a	39.43 ± 1.880 a	***
Feret min (µm)	33.79 ± 0.863 ab	36.44 ± 1.394 a	26.13 ± 1.260 c	30.51 ± 1.375 b	34.82 ±1.742 a	***
Aspect Ratio	1.271 ± 0.027 ab	1.196 ± 0.021 b	1.366 ± 0.065 a	1.334 ± 0.025 a	1.222 ± 0.022 b	**
Sphericity	0.668 ± 0.032 ab	0.731 ± 0.030 a	0.619 ± 0.051 b	0.586 ± 0.027 b	0.680 ± 0.029 ab	*
Convexity	0.942 ± 0.009 ab	0.949 ± 0.009 a	0.924 ± 0.013 c	0.930 ± 0.012 bc	0.946 ± 0.014 a	**
Starch (%)	0.782 ± 0.016 d	1.558 ± 0.021 cd	2.879 ± 0.012 c	12.93 ± 0.028 a	6.416 ± 0.013 b	***
Amyloplasts (n°)	7.730 ± 0.727 bc	5.780 ± 1.145 c	7.700 ± 1.356 bc	12.33 ± 0.702 a	10.73 ± 0.794 ab	***
Amyloplasts diameter (µm)	2.082 ± 0.116 d	2.872 ± 0.114 c	2.339 ± 0.197 d	5.236 ± 0.216 a	3.424 ± 0.180 b	***
TTCC (µm)	364.1 ± 8.804 b	407.7 ± 11.30 a	328.5 ± 11.94 c	357.7 ± 9.420 bc	421.2 ± 15.37 a	***
TA (µm)	79.83 ± 20.72 c	141.3 ± 42.86 bc	96.09 ± 25.47 c	263.0 ± 12.96 a	222.9 ± 42.28 ab	**
TA/TTCC Ratio	0.304 ± 0.064 c	0.702 ± 0.018 a	0.291 ± 0.079 c	0.734 ± 0.026 a	0.510 ± 0.084 b	***
**Intercellular Spaces**						
Area (µm)	13.73 ± 1.529 a	21.38 ± 4.501 a	17.28 ± 3.808 a	12.44 ± 1.683 a	12.54 ± 1.858 a	NS
Feret max (µm)	6.697 ± 0.546 a	8.082 ± 0.933 a	7.572 ± 1.268 a	6.409 ± 0.644 a	5.930 ± 0.391 a	NS
Feret mean (µm)	5.606 ± 0.424 a	6.646 ± 0.731 a	6.021 ± 0.921 a	5.290 ± 0.456 a	4.919 ± 0.355 a	NS
Feret min (µm)	4.199 ± 0.276 a	4.753 ± 0.441 a	3.729 ± 0.388 a	3.788 ± 0.258 a	3.576 ± 0.335 a	NS
Aspect Ratio	1.594 ± 0.059 a	1.687 ± 0.066 a	1.907 ± 0.159 a	1.677 ± 0.130 a	1.842 ± 0.168 a	NS
Sphericity	0.430 ± 0.038 a	0.402 ± 0.047 a	0.350 ± 0.054 a	0.414 ± 0.048 a	0.365 ± 0.052 a	NS
Convexity	0.824 ± 0.026 a	0.817 ± 0.027 a	0.864 ± 0.039 a	0.829 ± 0.030 a	0.873 ±0.025 a	NS

NS, *; **, and ***, Not significant or significant at *p* < 0.05, 0.01, and 0.001, respectively. Different letters within each column indicate significant differences according to Duncan multiple comparison tests (*p* ≤ 0.05).

**Table 3 plants-10-02272-t003:** Effect of irradiation on cell and intercellular spaces size (area and Feret maximum, mean, and minimum diameters), shape (aspect ratio, sphericity, and convexity), and starch content (starch%, number of amyloplasts, amyloplast diameter, total radius (TRS), thickness of the layers of cells containing the amyloplasts (TA), and their ratio (TA/TRS)) in hypocotyl stele of eye bean seedlings from control and seeds irradiated with C- and Ti-ions at 1 and 10 Gy doses. Mean values and standard errors are shown (*n* = 60 for cells/amyloplasts/intercellular spaces; *n* = 18 for tissue thickness and starch%).

Hypocotyls—Stele						
Parameters	Ct	1C	10C	1Ti	10Ti	S
**Parenchyma Cells**						
Area (µm)	1337 ± 125.1 a	1291 ± 72.56 a	1329 ± 124.6 a	1321 ± 114.7 a	1702 ± 171.6 a	NS
Feret max (µm)	48.04 ± 2.332 b	47.12 ± 1.362 b	51.75 ± 1.595 ab	47.27 ± 1.986 b	55.02 ± 2.614 a	*
Feret mean (µm)	42.85 ± 2.026 a	42.61 ± 1.186 a	44.26 ± 1.722 a	43.04 ± 1.806 a	48.68 ± 2.309 a	NS
Feret min (µm)	36.40 ± 1.730 a	38.24 ± 1.161a	35.51 ± 1.957 a	37.75 ± 1.767 a	41.34 ± 2.140 a	NS
Aspect Ratio	1.297 ± 0.031 b	1.220 ± 0.022 b	1.491 ± 0.065 a	1.248 ± 0.035 b	1.328 ± 0.035 b	***
Sphericity	0.642 ± 0.033 ab	0.726 ± 0.030 a	0.506 ± 0.041 c	0.677 ± 0.032 ab	0.586 ± 0.026 bc	***
Convexity	0.939 ± 0.008 a	0.943 ± 0.008 a	0.927 ± 0.010 a	0.932 ± 0.007 a	0.940 ± 0.018 a	NS
Starch (%)	1.598 ± 0.007 c	1.240 ± 0.019 c	2.708 ± 0.014 c	10.58 ± 0.033 a	5.581 ± 0.023 b	***
Amyloplasts (n°)	4.380 ± 0.584 b	5.080 ± 0.742 b	6.000 ± 1.006 b	12.57 ± 1.155 a	10.33 ± 1.103 a	***
Amyl. diameter (µm)	2.935 ± 0.144 c	2.662 ± 0.109 c	2.394 ± 0.136 c	4.383 ± 0.287 a	3.504 ± 0.204 b	***
TRS (µm)	283.2 ± 27.67 b	282.0 ± 11.97 b	279.7 ± 14.41 b	319.6 ± 11.12 ab	356.6 ± 13.03 a	*
TA (µm)	180.3 ± 38.67 bc	173.8 ± 38.44 bc	136.3 ± 34.33 c	275.8 ± 26.48 ab	326.3 ± 24.79 a	**
TA/TRS Ratio	0.715 ± 0.083 ab	0.826 ± 0.096 a	0.504 ± 0.126 b	0.862 ± 0.074 a	0.907 ± 0.048 a	*
**Intercellular Spaces**						
Area (µm)	20.03 ± 2.552 a	24.33 ± 5.209 a	22.90 ± 4.320 a	22.36 ± 3.965 a	31.84 ± 4.413 a	NS
Feret max (µm)	7.356 ± 0.574 a	7.764 ± 0.617 a	8.624 ± 1.070 a	8.920 ± 1.445 a	10.43 ± 0.834 a	NS
Feret mean (µm)	6.231 ± 0.460 a	6.576 ± 0.519 a	7.076 ± 0.828 a	7.232 ± 1.013 a	8.417 ± 0.626 a	NS
Feret min (µm)	4.640 ± 0.341 a	4.893 ± 0.366 a	5.020 ± 0.433 a	5.074 ± 0.436 a	5.712 ± 0.428 a	NS
Aspect Ratio	1.605 ± 0.083 a	1.610 ± 0.074 a	1.681 ± 0.073 a	1.641 ± 0.087 a	1.865 ± 0.138 a	NS
Sphericity	0.469 ± 0.050 a	0.465 ± 0.043 a	0.392 ± 0.043 a	0.395 ± 0.045 a	0.381 ± 0.052 a	NS
Convexity	0.870 ± 0.026 a	0.861 ± 0.021 a	0.815 ± 0.031 a	0.829 ± 0.033 a	0.809 ± 0.023 a	NS

NS, *; **, and ***, Not significant or significant at *p* < 0.05, 0.01, and 0.001, respectively. Different letters within each column indicate significant differences according to Duncan multiple comparison tests (*p* ≤ 0.05).

**Table 4 plants-10-02272-t004:** Effect of C- and Ti-ion irradiation on total chlorophylls (Chl a, b, and a + b) and carotenoids (x + c) content in seedlings of eye bean from control and seeds irradiated at 1 and 10 Gy doses. Mean values and standard errors are shown (*n* = 5).

Chlorophyll and Carotenoid Content						
Parameters	Ct	1C	10C	1Ti	10Ti	S
Chl a (mg/g)	1.504 ± 0.081 a	1.293 ± 0.036 b	0.701 ± 0.033 d	1.072 ± 0.030 c	1.278 ± 0.071 b	***
Chl b (mg/g)	0.652 ± 0.048 a	0.596 ± 0.068 a	0.353 ± 0.024 b	0.602 ± 0.021 a	0.548 ±0.035 a	**
Chl a + b (mg/g)	2.157 ± 0.117 a	1.889 ± 0.101 b	1.054 ± 0.024 c	1.673 ± 0.015 b	1.826 ± 0.095 b	***
x + c (mg/g)	0.342 ± 0.037 a	0.233 ± 0.024 b	0.128 ±0.008 c	0.231 ± 0.024 b	0.282 ± 0.032 ab	**

NS, *; **, and ***, Not significant or significant at *p* < 0.05, 0.01, and 0.001, respectively. Different letters within each column indicate significant differences according to Duncan multiple comparison tests (*p* ≤ 0.05).

## Data Availability

The data supporting the results of this study are accessible from the corresponding author, upon reasonable request.

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
