# Peer review of "Comparative Analysis of the Effect of Carbon- and Titanium-Ions Irradiation on Morpho-Anatomical and Biochemical Traits of Dolichos melanophthalmus DC. Seedlings Aimed to Space Exploration"

_plants, 2021, doi:10.3390/plants10112272_

Round 1

Reviewer 1 Report

The manuscript nicely represents the effect of high-LET ionizing radiation on eye bean seeds. The authors analyzed not only morphological changes but also biochemical traits after the irradiation in two different doses. Materials and methods section is also well described in details. Statistical analyses were done properly. Although it is not difficult to find similar works to the work done in this manuscript, the results from this manuscript will ramifiy knowledge on plant responses to radiations in Space.

I have few comments to be minorly corrected before publication.

1. Line 56: hydrogen peroxidase -> hydrogen peroxide

2. Line 348: remove "Three sets of 30 seeds per eac"

3. Line 410: 13.000 rpm -> 13,000 rpm

4. What does "Scheme 0. 0.01, and 0.001, respectively" in footnote below each table mean? 

Author Response

Reviewer 1

C6: The manuscript nicely represents the effect of high-LET ionizing radiation on eye bean seeds. The authors analyzed not only morphological changes but also biochemical traits after the irradiation in two different doses. Materials and methods section is also well described in details. Statistical analyses were done properly. Although it is not difficult to find similar works to the work done in this manuscript, the results from this manuscript will ramifiy knowledge on plant responses to radiations in Space. I have few comments to be minorly corrected before publication.
R6: We thank the reviewer 1 for assessing the quality of our manuscript.  We carefully accepted all his/her suggestions.

C7: Line 56: hydrogen peroxidase -> hydrogen peroxide
R7: Done (L 71).

C8: Line 348: remove "Three sets of 30 seeds per eac"
R8: Done (L 406).

C9: Line 410: 13.000 rpm -> 13,000 rpm
R9: We modified the rpm value in g-equivalent as requested by reviewer 4.

C10: What does "Scheme 0. 0.01, and 0.001, respectively" in footnote below each table mean? 
R10: We corrected the mistake in all tables’ footnote.

Reviewer 2 Report

The concept of the manuscript is interesting and the manuscript has been well-written. The figures and graphs are self-explanatory and the pictures are of good quality. All the concepts have been explained clearly. The experiment seems to be executed properly.

There are some questions and edits suggested:

Why did the authors specifically chose Dolichos melanophthalamus DC for this experiment?

Line 348: Please delete the words that are repeated.

Line 410: It is suppose to be 13,000 rpm and not 13.000 rpm.

Author Response

Reviewer 2

C11: The concept of the manuscript is interesting and the manuscript has been well-written. The figures and graphs are self-explanatory and the pictures are of good quality. All the concepts have been explained clearly. The experiment seems to be executed properly. There are some questions and edits suggested:
R11: We wish to thank reviewer 2 for his/her precious comments. We accepted all his/her suggestions.

C12: Why did the authors specifically chose Dolichos melanophthalamus DC for this experiment?
R12: We specified in materials and methods that this species was chosen for the high protein and aminoacids contents. We also added a reference on the nutritional value of the species. Please note that we also reported the synonym botanical name of this species. Done (L 381, 385-386).

C13: Line 348: Please delete the words that are repeated.
R13: Done (L 406).

C14: Line 410: It is suppose to be 13,000 rpm and not 13.000 rpm.
R14: Yes, the reviewer is right: it was a typo. Anyway, we modified the rpm value in g-equivalent as requested by reviewer 4.

Reviewer 3 Report

The subject of this study is very interesting and innovative, as well as necessary for the imminent long-duration space travel that will take place in the very near future.

However, I believe that the authors should have used more doses and ions to be able to argue with more confidence the hypotheses presented in the discussion. If this is not possible to do, at least an explanation should be presented as to why this was not done.

Author Response

Reviewer 3

C15: This paper describes in detail the effects of two types of ionizing radiation on the morphological characteristics of dried seeds of Dolichos melanophthalmus DC. (eye bean). It also describes, with less detail, the effects of these two types of high linear energy transfer radiation on the content of chlorophyll and carotenoids, as well as ascorbic acid and hydrogen peroxide, the latter as an indicator of oxidative stress.
The subject of this study is very interesting and innovative, as well as necessary for the imminent long-duration space travel that will take place in the very near future. However, I believe that the authors should have used more doses and ions to be able to argue with more confidence the hypotheses presented in the discussion. If this is not possible to do, at least an explanation should be presented as to why this was not done.
R15: We thank the reviewer 3 for his/her comments. We are aware that using more doses would have been really interesting and useful to discuss our hypothesis. However, this was not possible due to limitations in the beam-time allocated to our experiment. Therefore, we were obliged to choose only two doses. We added more information about the reasons why we chose Carbon and Titanium at 1 Gy and 10 Gy (L 54-67, 121-132)

C16: Line 2: Carbon (C) and Titanium (Ti)
R16: Done (L 2).

C17: Line 56: Hydrogen peroxide
R17: Done (L 71).

C18: Lines 49-52: Which of these radiations is the most harmful, and in what proportion can we find them in space? This is to know why you have decided to use high-LET ionizing radiation.
R18: Ti-ions have higher LET than C-ions, therefore Ti-ions should be more harmful since they transfer more energy to the material traversed per unit distance. However, this phenomenon could be compensated by the higher Z-number of Ti that should correspond to lower relative biological effectiveness for some biological parameters (Jones and Hill, 2020 Biomedical Physics & Engineering Express6(5), 055001.). Unfortunately, available information is very scattered, and more experiments are needed to have a clearer understanding especially on plant models.  We cannot be more precise on the proportion of these ions in space radiation, because of their high spatial and temporal variability. However, experiments irradiating plants with C and Ti ions are relevant for preliminary experiments aimed at Space exploration given they are among ions considered proxy to galactic nuclei and are the most used to simulate the GRC spectrum in irradiation facilities on Earth. We added relevant information and related references in the text (lines 54-67).

C19: Lines 68-70: The diversity of plant tissues is very large, and certainly different between animals and plants, which is why perhaps this paragraph should be more specific, for example in plant seeds.
R19: Done (L 84-85).

C20: Lines 71-73: It is not very clear to me how this paragraph relates to the previous one. Perhaps including the name of the ions (C and Ti) could make it more precise.
R21: Done (L 86).

C21: Line 101: Why this species? Fast germination, high nutritional content?
R21: We added more information on the species. This species was chosen for the high protein and amino acids contents. We also added a reference on the nutritional value of the species. Please note that we also reported the synonym botanical name of this species. (L 381, 385-386).

C22: Figure 1 (a): Please set the ordinate axis to 100%.
R22: Done (Figure 1a).

C23: Line 115: Please add the number of samples analyzed.
R23: We added requested information.

C24: Line 120: What is this typical structure?
R24: We changed the word typical (L 147).

C25: Line 127: Please try to be more specific: “more irregular”
R25: We modified this sentence accordingly (L 154-155).

C26: Line 137: It would be helpful to add a diagram showing the different tissues shown in the tables, as well as the parameters being measured. This would probably be in the materials and methods section.
R26: We added a diagram in the material and method section.

C27: Lines 141-142: Idem to line 115.
R27: Done.

C28: Figure 2: Is it possible to present all figures at the same magnification level?
R28: We prefer keeping these magnifications. We purposely chose the magnification levels more suitable in order to better represent details of specific anatomical/cytological features described.

C29: Figure 3: Idem to figure 2.
R29: Please refer to R28.

C30: Figure 4: Idem to line 115
R30: Done.

C31: Lines 234-236: This argument is based on only two doses and two ions, so it is difficult to assert that germination responses are ion and dose specific.
R31: We rephrased the sentence. (L 271-272).

C32: Lines 239-241: As in the previous case, only a few tissues were analyzed, so it is difficult to take this idea for granted.
R32:  We rephrased the sentence. (L 273-274).

C33: Lines 252-253: What could be the reason for this?
R33: This might be due to the fact that in actively growing tissues, the process by which cell wall loosening (which induces wall stress relaxation which in turn generates the reduced water potential that is needed for water uptake and volumetric expansion of the cell) is active while cell walls are still being deposited and  cells would enlarge much more than in already formed tissues (as cotyledons in Fabaceae) in which cell wall loosening would only cause a slight enlargement of the protoplasts because cell wall itself is already mature and there is higher mechanical constraint due to not growing surrounding cells. We summarized this phenomenon in the text (L 289-292).

C34: Line 276: its possible
R34: Done (L 317).

C35: Lines 279-282: Conversion is a process controlled by phytohormones? Could radiation have any effect on some of them?
R35: We agree with the reviewer that phytohormones could play a role in the control of many radiation-induced responses. However, since we did not measure hormones, we prefer not to speculate on this.

C36: Lines 291-293: At what point was the H2O2 content determined?
R36: the H2O2 was determined at the end of the experiment at the harvesting. We added more information in the material and methods section (L 416-417, 457). 

C37: Line 335: Why were only two doses used? In my opinion this greatly limits the hypotheses that have been developed in the discussion. It is possible that some morphological responses may have biphasic or triphasic behavior, but this cannot be determined with only two of them.
R37: We thank the reviewer 3 for his/her comments. We are aware that using more doses would have been really interesting and useful to discuss our hypothesis. However, this was not possible due to limitations in the beam-time allocated to our experiment. Therefore, we were obliged to choose only two doses. We added more information about the reasons why we choose Carbon and Titanium at 1 Gy and 10 Gy  (L 54-67).

C38: Lines 341-342: Were the seeds protected from light during the transfer?
R38: Yes, we added a clarification (L 393).

C39: Line 348: Please delete “Three sets of 30 seeds per each”
R39: Done (L 406).

C40: Line 348: Where were the seeds obtained?
R40: We added this information (L 382).

C41: Lines 357-358: Please read line 137.
R41: Done.

C42: Lines 380-383: Is this done by using a software?
R42: Yes, AnalySIS 3.2 (Olympus), as specified in Line 430.

Reviewer 4 Report

The authors had legume seeds irradiated and present mostly morphological data on plants cultivated for 10 days.

The problems with the paper are that the radiation (C and Ti) is not comparable to the dominant radiation source (i.e., protons) that are encountered during space transit. The justification for these ions is not based on biological effects (breeding programs are mutagenesis-based) and exposure of dry seeds does not lead to radiolysis. As such the text (P2, L65 – 82) is not a justification for the intended work. In addition, the methodology is poorly described as the treatment and ion fluence conditions are not provided. Claiming ‘larger berries’, a clearly genetically influenced feature as example for vegetative assessments is misguided.

The authors do not discuss why a more massive ion (Ti) affects processes past germination less or have hormetic effects (Fig. 1b) than less massive ions (C). The lack of a proper analysis (altered fluence rates, more Gy levels, exposure before and after imbibition etc.) make this study irrelevant for radiation bio0logy. The provided morphometric data may be of use but comparisons (other legumes, seed size, age) would be necessary.

Technical aspects

Were seeds exposed in a single layer or in bulk possibly providing shielding? :

What is the purpose of listing ranges as less than (L66, 67)?

What is the meaning of dangerous? LET depends on the energy of the ions (i.e., Bragg peak), not just their mass.

Plants reacting to abiotic stress is an incorrect view of seeds irradiated in their dry state.

ROS is the short-term response to all types of stresses; a continuously elevated ROS signal, which could be correlated to irradiation,  is not provided.

Although an explanation of the term ‘Feret’ was attempted, it is unclear is a transverse or longitudinal section was used; this term is unnecessarily confusing.

L 401:10,000 rpm means nothing unless the radius of the rotor is provided; report as g- equivalents.

No information is provided on the thickness of the sections. Sections thinner than a structure may underestimate its true dimension, thick sections lead to overlaps and poor resolution.

What I the purpose of listing sphericity and convexity? Are they indicative of sensitive to embedding artefacts?

The uneven distribution of amyloplasts (sedimented in endodermal cell, possibly during fixation) make starch determinations questionable (what is starch percentage; what would be 100%?).

What is meant by ‘Scheme 0’ (Table subscript)?

The identification of the micrographs (Fig. 2) is inadequate for spatial and organ definition.

Image A is of different magnification? (not included in caption)

Fig. 2H is likely of a different cut plane than the other panel in the center column.

Why are the images labeled in lower case letters but referred to with upper case?

Fig 3: based on the starch size (see note  earlier on section thickness) h and i shows much larger amyloplasts then the other panels in the two right columns. Therefore, the magnification must be different from b -o.

Fig. 4: the almost identical content of H2O2 and Vitamin C is suspicious as oxidative stress (ROS/H2O2) is inversely proportional to reduction equivalents (Vit. C).

 Chlorophyll quantities are the result of acute light effects (fluence, day-length and Mg/N/Fe (mineral) content) and their variability has not been considered .

The phrase ‘The increased size of cells in cotyledons of irradiated seeds was likely favored by a possible radiation-induced loosening of cell walls’ is purely speculative and not supported by any actual or deduced argument.

The claimed ion-specificity is not tenable as only C ions showed effects, but other ions were not studied at different energies, times, or fluences.

Unfortunately, a host of other statements are dubious, not related to data (either in this paper or the literature) and often taken out of context.

Author Response

Reviewer 4

C43: The authors had legume seeds irradiated and present mostly morphological data on plants cultivated for 10 days. The problems with the paper are that the radiation (C and Ti) is not comparable to the dominant radiation source (i.e., protons) that are encountered during space transit. The justification for these ions is not based on biological effects (breeding programs are mutagenesis-based) and exposure of dry seeds does not lead to radiolysis. As such the text (P2, L65 – 82) is not a justification for the intended work. In addition, the methodology is poorly described as the treatment and ion fluence conditions are not provided. Claiming ‘larger berries’, a clearly genetically influenced feature as example for vegetative assessments is misguided.
R43: We agree with the reviewer that C- and Ti-ions alone cannot simulate the whole GCR; however, it still remains that experiments with acute doses of single ions are preliminary to Space experiments. Among the heavy ions, carbon and titanium are the most used to simulate on Earth the GRC spectrum in irradiation facilities on Earth ( Simonsen et al., 2020;Schuy et al., 2020NASA,). C-ions irradiation are more used than Ti-ions in Space oriented experiments of plant biology. Fe and Ti ions are used as proxy to galactic nuclei (Lobascio et al., 10.1097/01.HP.0000288560.21906.4e ). We added available information in the text to justify the chosen ions and doses. We also added details about the irradiation procedure and relevant information to calculate fluence if needed (L 54-67, 121-132, 389-392).

We also checked the sentence in which we report on the effects of irradiation on tomato and we confirm we reported it only as example of radiation induced modifications and not of course as an example of vegetative assessment.

C44: The authors do not discuss why a more massive ion (Ti) affects processes past germination less or have hormetic effects (Fig. 1b) than less massive ions (C). The lack of a proper analysis (altered fluence rates, more Gy levels, exposure before and after imbibition etc.) make this study irrelevant for radiation bio0logy. The provided morphometric data may be of use but comparisons (other legumes, seed size, age) would be necessary.
R44:  We thank the reviewer 3 for his/her comments. We are aware that using more doses, irradiation on several target stages and species, would have been really interesting and useful to discuss our hypothesis. However, this was not possible due to limitations in the beam-time allocated to our experiment. Therefore, we were obliged to choose only two doses. We added more information about the reasons why we chose Carbon and Titanium at 1 Gy and 10 Gy (L 54-67, 121-132). Ti-ions used have higher LET than C-ions, therefore Ti-ions should be more harmful since they transfer more energy to the material traversed per unit distance. However, this phenomenon could be compensated by the higher Z-number of Ti that is reported to correspond to lower relative biological effectiveness for some biological parameters. Unfortunately, available information is very scattered and we prefer reporting our results discussing the biological meaning but not speculating on possible mechanisms of interaction among different ions and cells.

C45: Were seeds exposed in a single layer or in bulk possibly providing shielding?
R45: Seeds were exposed in a few layers that were not enough to provide shielding.  

C46: What is the purpose of listing ranges as less than (L66, 67)?
R46: We modified the sentence following reviewer 4 suggestions (L 81).

C47: What is the meaning of dangerous? LET depends on the energy of the ions (i.e., Bragg peak), not just their mass.
R47: We rephrased the text (L 85).

C48: Plants reacting to abiotic stress is an incorrect view of seeds irradiated in their dry state.
R48: We agree with this comment, but here we are discussing the role of starch in seedlings and not in irradiated dry seeds. We added some clarification about it. (L 110-111).

C49: ROS is the short-term response to all types of stresses; a continuously elevated ROS signal, which could be correlated to irradiation, is not provided.
R49: if the reviewer 4 refer to line 113-117, we referred to the accumulation of ROS due to radiation as reported in Ahuja et al [39].

C49: Although an explanation of the term ‘Feret’ was attempted, it is unclear is a transverse or longitudinal section was used; this term is unnecessarily confusing.
R49: The term “Feret” is commonly used in digital image analysis to quantify the size and shape of cells in tissue sections. We already reported that we analyzed cross sections in materials and methods; however,  we added again this information also in the description of the index (L 438).

C50: L 401:10,000 rpm means nothing unless the radius of the rotor is provided; report as g- equivalents.
R50: We added the g-equivalents values (L 470,477).

C51: No information is provided on the thickness of the sections. Sections thinner than a structure may underestimate its true dimension, thick sections lead to overlaps and poor resolution.
R51: This information is reported at line 422.

C52: What I the purpose of listing sphericity and convexity? Are they indicative of sensitive to embedding artefacts?
R52: Sphericity and convexity are morpho-anatomical parameters commonly used in morpho-anatomical studies to characterize cell shape. The embedding procedure was the same for all treatments, so differences in cell size and shape are not ascribed to artifacts. We added information to clarify the text (L 435).

C53: The uneven distribution of amyloplasts (sedimented in endodermal cell, possibly during fixation) make starch determinations questionable (what is starch percentage; what would be 100%?).
R53:  With starch percentage we refer to the % of tissue occupied by the amyloplast as reported in lines 448-452. Though, the 100% would be if the whole tissue was occupied by the amyloplasts. We did not measure starch in endodermal cells.

C54: What is meant by ‘Scheme 0’ (Table subscript)?
R54: There was a typo. We corrected it in all the table subscript.

C55: The identification of the micrographs (Fig. 2) is inadequate for spatial and organ definition.
R55: We added a scheme in the methods to clarify the position of the analyzed tissues.

C56: Image A is of different magnification? (not included in caption)
R56: In both figures, images (a,d,g,j,m) are all at the same magnification with bars = 100µm, while images (b,e,h,k,n,c,f,I,l,o) are all at the same magnification with bars = 100 µm, as reported in the caption.

C57: Fig. 2H is likely of a different cut plane than the other panel in the center column.
R57: We agree and chose a better image for the 2H.

C58: Why are the images labeled in lower case letters but referred to with upper case?
R58: We corrected this mistake.

C60: Fig 3: based on the starch size (see note earlier on section thickness) h and i shows much larger amyloplasts then the other panels in the two right columns. Therefore, the magnification must be different from b -o.
R60: We’r sorry, it was a mistake: we inverted some panels in the final editing of the picture. From b-o the magnification is the same but different compared to a- m. Please also see R52.

C61: Fig. 4: the almost identical content of H2O2 and Vitamin C is suspicious as oxidative stress (ROS/H2O2) is inversely proportional to reduction equivalents (Vit. C).
R61: We are only in part in agreement with this statement. Generally, the plant response to oxidative stress may include an inversely proportional relationship between ROS occurrence and antioxidants (in our case H2O2 vs AsA) but may also determine an increase in antioxidant compounds engaged in scavenging the free radicals. Thus, it is not surprising that endogenous H2O2 and AsA move parallelly, showing the same trend (Dipierroet al 2005, journal of Plant Physiology 162 (2005) 529—536).
Moreover, the H2O2 increase may also act as a signal in triggering the antioxidant response in the plant.

C62: Chlorophyll quantities are the result of acute light effects (fluence, day-length and Mg/N/Fe (mineral) content) and their variability has not been considered .
R62: Their variability has not been considered since the seedlings were grown and sampled in the same environmental conditions.

C63: The phrase ‘The increased size of cells in cotyledons of irradiated seeds was likely favored by a possible radiation-induced loosening of cell walls’ is purely speculative and not supported by any actual or deduced argument.
R63: We smoothed the sentence but preferred to keep it also because the other reviewers asked to add more comments on it (L 280).

C64: The claimed ion-specificity is not tenable as only C ions showed effects, but other ions were not studied at different energies, times, or fluences.
R64: We rephrased the sentence (L 355).

C65: Unfortunately, a host of other statements are dubious, not related to data (either in this paper or the literature) and often taken out of context.
R65:  We smoothed some comments to avoid speculations. 

Round 2

Reviewer 3 Report

The fact that only two doses were studied remains the major limitation of the study, so the authors should in my opinion mention that this needs to be expanded in future research.

Author Response

Reviewer 2

C5: The fact that only two doses were studied remains the major limitation of the study, so the authors should in my opinion mention that this needs to be expanded in future research.

R5: We added more about it in lines 359-360.